# Asparagine: A Metabolite to Be Targeted in Cancers

**DOI:** 10.3390/metabo11060402

**Published:** 2021-06-19

**Authors:** Jie Jiang, Sandeep Batra, Ji Zhang

**Affiliations:** 1Herman B Wells Center for Pediatric Research, School of Medicine, Indiana University, Indianapolis, IN 46202, USA; jj15@iu.edu; 2Riley Hospital for Children at Indiana University Health; Indianapolis, IN 46202, USA; 3Department of Biochemistry and Molecular Biology, School of Medicine, Indiana University; Indianapolis, IN 46202, USA

**Keywords:** asparagine, L-asparaginase, acute lymphoblastic leukemia, asparagine synthetase, stress response, metabolic adaptation, mTORC1, GCN2, ATF4

## Abstract

Amino acids play central roles in cancer progression beyond their function as building blocks for protein synthesis. Thus, targeting amino acid acquisition and utilization has been proved to be therapeutically beneficial in various pre-clinical models. In this regard, depletion of circulating asparagine, a nonessential amino acid, by L-asparaginase has been used in treating pediatric acute lymphoblastic leukemia (ALL) for decades. Of interest, unlike most solid tumor cells, ALL cells lack the ability to synthesize their own asparagine de novo effectively. However, only until recently, growing evidence suggests that solid tumor cells strive to acquire adequate amounts of asparagine to support tumor progression. This process is subjected to the regulation at various levels, including oncogenic signal, tumor-niche interaction, intratumor heterogeneity and dietary accessibility. We will review the literature on L-asparaginase-based therapy as well as recent understanding of asparagine metabolism in solid tumor progression, with the hope of shedding light into a broader cancer therapeutic strategy by perturbing its acquisition and utilization.

## 1. Introduction

Amino acids are the fundamental building blocks for the synthesis of protein, which contributes to the majority of biomass in proliferating mammalian cells [1]. Growing evidence suggests that the demands for amino acids in proliferating mammalian cells, such as cancer cells, go beyond their requirement for global protein synthesis [2]. As a result, restricting amino acid acquisition and utilization has been proposed to be a potential therapeutic strategy to limit cancer cell growth while leaving the normal tissues largely intact. A compelling example of such therapy is the application of L-asparaginase to deplete circulating asparagine in treating pediatric ALL patients for decades [3]. Despite the successful application in ALL patients, L-asparaginase has not been proved to be effective in most other cancer types, suggesting their reduced dependency on circulating asparagine. However, recent advancements in cancer cell metabolism suggest that asparagine plays critical roles in solid tumor progression, and is therapeutically explorable. In this review, we will summarize the application of L-asparaginase in ALL, discuss the potential mechanisms driving therapeutic resistance, and highlight the most recent studies elucidating the role of asparagine as a nutrient or signaling modulator to support solid tumor progression, and discuss therapeutic implications.

## 2. Asparagine and L-Asparaginase in Acute Lymphoblastic Leukemia (ALL)

### 2.1. History of L-Asparaginase

L-asparaginase was first discovered in 1953 by two studies that identified anti-tumor activity of guinea pig serum toward transplanted lymphomas [4,5]. Later on, several studies confirmed that the L-asparaginase activity found in the guinea pig serum is responsible for this anti-tumor effect [6,7,8,9]. Native L-asparaginase purified from *E. coli* was first used to induce remission in ALL in the 1960s [10], and ever since has been used extensively in patients with ALL and acute lymphoblastic lymphoma (ALLy) [3,11]. It exerts its metabolic effect by catabolizing asparagine to aspartate and free ammonium in the circulation [3]. In 1980s, chemically modified *E. coli* L-asparaginase was developed using a polyethylene glycol moiety conjugation, also known as pegylated L-asparaginase (PEG-L-asparaginase) [12]. This chemical modification significantly prolongs its half-life and reduces antigenicity, but with similar pharmacokinetics (PK) [13,14]. Asparaginase can also be derived from Erwinia chrysanthemi. In patients with severe allergic reactions to native *E. coli* L-asparaginase or PEG-L-asparaginase, Erwinia asparaginase is often substituted [15,16,17], as antibodies to native or PEG-L-asparaginase, usually do not cross react with the Erwinia-derived asparaginase. However, Erwinia asparaginase has a short half-life and has to be administered every 48–72 h [14]. While early studies demonstrated inferior outcomes with its usage, the dose frequency might have been suboptimal and lead to these results [18]. Indeed, recent studies have demonstrated adequate depletion with frequent administration of Erwinia asparaginase [16,19]. More recently, a new modified form, EZN-2285 (SC-PEG *E. coli* L-asparaginase or Cal-asparaginase) is under investigation, and will likely replace PEG-L-asparaginase use. EZN-2285 is produced by replacing the succinimidyl succinate (SS) linker in PEG-L-asparaginase with a succinimidyl carbamate (SC) linker, thereby leading to a more hydrolytic stability, and decreased susceptibility to hydrolytic removal of the PEG from the protein conjugate [20].

L-asparaginase is included in almost all current regimen for pediatric ALL/ALLy and ALLy due to its unique efficacy toward ALL/ALLy blasts. However, little is known on why other cancers are resistant to it [10,21]. One explanation is that compared to other cancer cells, ALL blasts express low levels of asparagine synthetase (ASNS) due to DNA hypermethylation on the gene promoter, and therefore rely on exogenous asparagine completely [22,23]. Metabolomic studies with genome-wide DNA methylation landscaping identified a small group of gastric and hepatic cancer cells that have similar features to ALL cells, raising the possibility to treat these cancers with L-asparaginase. Another interesting question is to develop humanized L-asparaginase to minimize antigenicity. Although evidence exists to support the possibility [24], mammals do not express functional L-asparaginase endogenously. In this regard, why only guinea pig serum contains L-asparaginase activity is still a mystery, since earlier studies using rabbit or horse serum did not identify anti-tumor activities [4]. Furthermore, when a functional L-asparaginase derived from lower organism is engineered into mammalian cells, it prevents mammalian cells from adapting to glutamine starvation [25]. These results suggest that loss of L-asparaginase activity during evolution may reflect a selective strategy to adapt to nutrient availability in the context of development. 

### 2.2. Clinical Responses to L-Asparaginase Treatment in ALL Patient

Since its first application in patient, many clinical studies have been performed to optimize the usage of L-asparaginase in treating ALL. The Children’s Cancer group (CCG) conducted a pivotal randomized clinical trial (CCG-1962) that established the effectiveness of PEG-L-asparaginase by comparing native to PEG-L-asparaginase in a randomized fashion [26]. PEG-L-asparaginase was given intramuscularly at 2500 IU/m^2^, single dose during induction and once during each delayed intensification phase. *E. coli* L-asparaginase (native form) was administered intramuscularly at a standard dose of 6000 IU/m^2^, three times weekly for 9 doses during induction and for 6 doses during each delayed intensification phase [26]. It was observed that patients who received PEG-L-asparaginase cleared their blasts more rapidly, had lower levels of detectable antibodies, and asparaginase levels persisted for a longer duration, compared to patients in the native arm of the study [26]. 

Frontline multi-institutional CCG studies (1941, 1962 and 1961) further established the ideal PK parameters for PEG-L-asparaginase, including adequate asparagine depletion ≤ 3 µM achieved with serum asparaginase levels ≥ 0.1 IU/mL. These studies also helped establish PK-based optimal dosing for Erwinia asparaginase (25,000 IU/m^2^, every 2–3 days) [27]. Dana-Faber Cancer Center reported that intravenous administration of PEG-L-asparaginase was safe and was similar to intramuscularly in children, with persistent and therapeutic serum asparaginase concentrations and minimal side-effects [28]. These results were confirmed in adult trials with pharmacodynamics (PD) demonstrating therapeutic enzymatic activity of 0.1 IU/mL or more for at least 3 weeks [29]. Of interest, L-asparaginase from *Pseudomonas 7A* contains glutaminase activity with a higher Km toward glutamine as a substrate [30]. Although glutaminase activity was initially reported to be involved in effective asparagine depletion in ALL patients [31], other studies suggest that the glutaminase activity may not be central for its therapeutic effect at least in pre-clinical models in vivo [32,33,34].

Dose intensification of asparaginase therapy was investigated in a number of trials, however these resulted in increased side effects such as thrombosis, pancreatitis, severe hypoalbuminemia, and cachexia [35,36,37]. Recent reports also have suggested that highly intensified asparaginase treatment worsens the toxicity of other agents [38,39]. However, high risk B-precursor ALL patients receiving augmented Berlin-Frankfurt-Munster (BFM) therapy that includes 2–10 additional doses of PEG-L-asparaginase during post-Induction Intensification phases demonstrate improved outcomes [40]. Furthermore, patients with T-ALL who received intensified asparaginase also had improved outcomes on Pediatric Oncology Group (POG) trials [41,42]. The UKALL 2003 trial intensified PEG-L-asparaginase dosing (using a modified COG/BFM regimen) in standard and intermediate risk ALL patients, who were minimal residual disease (MRD) positive [36,37].

In summary, the clinical effect of L-asparaginase is determined by the formulations as well as the dose, route of administration and frequency of the treatment. In general, intensified regimen improves overall outcome but also introduces toxicities and side effects. Allergic reaction is one major side effect, which needs to be monitored closely but can be overcome with using substitutional approaches. 

### 2.3. Mechanism of Resistance to L-Asparaginase Treatment in ALL

Similar to many other chemo-agents, another factor to limit the clinical efficacy of L-asparaginase is the potential development of resistance. As mentioned earlier, the development of allergic reactions post-treatment is one mechanism seen commonly with its use. These IgG-mediated allergic reactions are unpredictable, idiosyncratic and can vary greatly in severity, and clinical symptoms, and can range from a minor rash or urticaria or fever to severe anaphylactic reactions to none [43,44]. Even without a symptom, the high titers of IgG production can lead to a rapid clearance of L-asparaginase, known as silent inactivation. Indeed, both missed dosing and silent inactivation has been linked to a greater risk of ALL relapse [18,45]. In addition, macrophage-mediated phagocytotic process can also contribute to the clearance of L-asparaginase in vivo at least in a mouse model [46]. 

The molecular and cellular mechanisms that cause resistance to L-asparaginase treatment have been extensively studied in cell culture experiments. A key player in driving the resistance is the expression of asparagine synthetase (ASNS) [3,47], the rate limiting enzyme for de novo biosynthesis of asparagine (Figure 1A). Earlier studies using cell culture showed that the expression levels of ASNS in human leukemic cell lines correlate reversely with their sensitivity to L-asparaginase treatment in vitro and forced expression of ASNS is sufficient to confer resistance in sensitive leukemic lines [48]. Additional studies suggested that the lack of expression of ASNS in sensitive leukemic cells is due to the DNA hypermethylation in the promoter region of *ASNS* gene [22,23,49]. Furthermore, recent work from our lab and others have shown that the capacity of ALL cells to induce the expression of *ASNS* following L-asparaginase treatment is a key determinant conferring therapeutic resistance [50,51]. This induction requires two components: (a) the general control nonderepressible 2 (GCN2) to activate ATF4, an indispensable transcriptional factor for *ASNS* transcription [52], and (b) the promoter demethylation of *ASNS* gene, which allows ATF4 recruitment and transactivation [51] (Figure 1B). Of interest, recent work suggests that Zinc Finger and BTB domain-containing protein 1 (ZBTB1) is required for ATF4-dependent transcription of *ASNS* gene specifically in T-ALL cells through promoter occupancy [53]. In addition to the expression of ASNS itself, studies suggest that the availability of aspartate and glutamine, two indispensable substrates of ASNS, are also important for cellular adaptation to the depletion of exogenous asparagine [54,55]. However, in cell culture conditions, glutamine uptake is not a limiting factor and most cancer cells can use glutamine to synthesize aspartate de novo [56]. Therefore, it warrants further investigation whether there is a cell type specific mechanism, such as aspartate and/or glutamine availability contributes to the sensitivity to L-asparaginase treatment in cells that express high levels of ASNS [54,55]. 

In contrast to the in vitro studies done in cell culture, contradictory conclusions were reached from clinical studies. Stams, et.al., showed that in ALL patients with TEL/AML1 fusion, despite that fact that a consistent increase in ASNS mRNA following L-asparaginase treatment was observed, no correlation was found in overall therapeutic responses [57]. However, this study was done in ALL patients with TEL/AML1 fusion, which is associated with good prognosis and high sensitivity to L-asparaginase in general. In addition, Su, et.al. reported that the level of ASNS protein is a better predictor of resistance to L-asparaginase treatment than mRNA [58]. However, a recent study with a larger cohort of T-ALL patient suggests that TLX1 positive T-ALL patients express low levels of ASNS mRNA and are more sensitive to L-asparaginase treatment [59]. Furthermore, the low expression of ASNS mRNA correlates with DNA hypermethylation in the promoter region of *ASNS* gene [59]. This study highlights the possibility to use ASNS mRNA expression and/or promoter methylation status of *ASNS* gene as a predictor for therapeutic responses. Thus, a larger cohort study including ALL patients that are more refractory to L-asparaginase treatment is needed. These studies also suggest that the genetic background of patients needs to be considered when interpreting the clinical results on therapeutic response/resistance. 

In addition to the intrinsic effect of ASNS expression in ALL cells, microenvironment can also contribute to L-asparaginase resistance. Iwamoto et.al, reported that bone marrow-derived mesenchymal cells (MSCs) express high levels of ASNS and thus can secrete asparagine to confer L-asparaginase resistance when ALL cells were co-cultured with MSCs [60]. Furthermore, bone marrow adipocytes can also protect ALL cells from L-asparaginase-induced cytotoxicity even though the mechanism is dependent on glutamine secretion [61]. In this study, obese children diagnosed with high-risk ALL has been found to have an increased risk of relapse than their lean counterparts [61]. Whether it reflects a mitigation of the glutaminase activity within L-asparaginase or represents an adaptive mechanism for ALL cells to use glutamine to drive de novo biosynthesis of asparagine warrants further investigation. 

Although most studies on the molecular mechanisms of L-asparaginase resistance have been focused on the expression of ASNS, other mechanisms can be involved. Using a genome-wide synthetic lethal CRISPR/Cas9 screen, activation of WNT pathway was identified to synergize with L-asparaginase treatment in inducing cell death in ALL cells that are resistant L-asparaginase treatment alone [62]. Mechanistically, WNT pathway suppresses GSK3-dependent proteolysis, which catabolizes unwanted cellular proteins as a scavenging pathway to maintain intracellular asparagine levels (Figure 2). However, in this study, whether the WNT/GSK pathway interferes with the expression of ASNS was not investigated. Similarly, autophagy is another mechanism for amino acid scavenging [1,63]. Along this line, activation of autophagy by L-asparaginase treatment has been found to protect ALL cells from death, even though it is still unclear whether intracellular asparagine levels can be restored through this mechanism or not [64] (Figure 2). Another study using genome-wide RNAi screen identified huntingtin associated protein 1 (HAP1) as a biomarker and a driver for the sensitivity to L-asparaginase treatment [65]. As a result, loss of HAP1 confers resistance to L-asparaginase treatment through preventing the release of calcium from endoplasmic reticulum (ER) and the subsequent calcium-dependent cell death [65] (Figure 2). Indeed, pharmacogenetic and functional genomic approaches have been used to identified genes associated with L-asparaginase resistance/sensitivity in ALL patient samples or cell lines [66,67,68,69,70]. However, most of these genes have not been broadly studied on their cellular functions in drug responses. 

In summary, the expression of ASNS in ALL cells particularly following L-asparaginase treatment is a key factor to drive therapeutic resistance. Amino acid scavenging through proteolysis and autophagy or inhibition of downstream signaling cascade, such as calcium flux, can also contribute the process. However, whether ASNS expression is a clinical predictor for therapeutic response is context-dependent and warrants further investigation. 

## 3. The Role of Asparagine in Other Types of Cancer

Even though clinical application of L-asparaginase is limited to ALL and some types of NK/T cell lymphoma, growing evidence suggests that asparagine bioavailability can play a critical role in the progression of other types of cancer. Along this line, various pre-clinical models have demonstrated the potential to combine L-asparaginase with other treatment or to restrict dietary asparagine as a means to treat cancer (Table 1). 

### 3.1. Asparagine in Promoting Solid Tumor Progression

The initial attention to asparagine in solid tumor progression was drawn by a seminal work showing that extracellular supplementation of asparagine can support tumor cell survival when exogenous glutamine is depleted [71]. Later on, similar phenomenon was observed in other types of proliferating mammalian cells [72,73], indicating a broader impact of the discovery. Since glutamine is often found to be limited in the tumor microenvironment [74], these studies provided the foundational basis in targeting asparagine bioavailability to prevent tumor cell adaptation to the lack of environmental glutamine. Mechanistically, asparagine prevents glutamine-depletion-induced ER stress in brain tumor cells [71]. Interestingly, a follow-up study from the same group found that asparagine can even support epithelial breast cancer cells to proliferate in the absence of exogenous glutamine [25]. In this context, asparagine supports de novo biosynthesis of glutamine through enhancing the expression of glutamine synthetase (GLUL) [25]. 

**Table 1 metabolites-11-00402-t001:** Summary of the role of asparagine in solid tumor studies.

References	Biological Processes	Functions
Zhang J, [71]	Glutamine starvation	Suppresses ER stress and apoptosis
Pavlova NN, [25]	Glutamine starvation	Supports GLUL expression and glutamine biosynthesis
Gwinn DM, [75]	KRAS-driven lung cancer	NRF2-dependent de novo biosynthesis to support tumor cell growth
LeBoeuf SE, [76]	KRAS-driven lung cancer	Demand for uptake to mitigate NRF2-dependent glutamate export
Linares JF, [77]	Prostate cancer	Secreted by CAFs to support tumor cell growth
Knott SRV, [78]	Breast cancer metastasis	Supports lung metastasis via EMT gene expression
Halbrook CJ, [79]	Pancreatic cancer	Protect tumor cells from ETC inhibition
Hinze L, [80]	Colorectal cancer	GSK3-dependent proteolytic scavenging to protect from L-asparaginase treatment

Recent work in a non-small-cell lung cancer (NSCLC) model shows that oncogenic KRAS signaling can hijack asparagine biosynthesis to support tumor growth [75]. Mechanistically, KRAS activates NRF2, a key transcription factor for anti-oxidant defense, through the downstream PI3K/AKT pathway. As a result, NRF2 induces the expression of ATF4 transcription factor to drive the transcription of *ASNS* gene. Of interest, the manuscript shows that AKT inhibitors can synergistically inhibit tumor growth in mice when combined with L-asparaginase [75]. In a separate study, KEAP1 mutant NSCLC cells were found to exhibit increased dependency on asparagine as well as several other nonessential amino acids (NEAAs) [76]. Since KEAP1 is a negative regulator of NRF2, the study demonstrated that enhanced NRF2 activity drives the import of cystine at the expense of intracellular glutamate, which is a key intermediate for the biosynthesis of NEAAs, including asparagine. As a result, L-asparaginase treatment or dietary restriction on asparagine can selectively inhibit the growth of KEAP1 mutant NSCLC cells in xenograft tumor models. However, further investigation is needed to reconcile whether NRF2 activation creates dependency on exogenous asparagine or dependency on its de novo biosynthesis. 

Solid tumor cells can also acquire asparagine from their microenvironment via stroma cell release. In a prostate cancer model, cancer associated fibroblasts (CAFs) can enhance asparagine biosynthesis and release asparagine to support tumor cell growth [77]. In this model, autophagy adaptor protein p62 is a key regulator to repress the expression of ATF4 through ubiquitin-mediated degradation. As a result, loss of p62 in the CAFs supports prostate cancer growth in a mouse model through ATF4-ASNS axis, likely by facilitating tumor cell adaptation to glutamine-limiting environment. However, glutamine is an indispensable precursor for asparagine biosynthesis. Why CAFs, but not tumor cells themselves, can produce asparagine in a glutamine-limiting environment warrants further investigation. 

Asparagine bioavailability can also affect tumor progression during specific stages. A recent work done in a metastatic breast cancer murine model shows that shRNA inhibition of ASNS expression selectively prevents lung metastasis while not perturbing tumor growth in the primary sites [78]. Mice treated with L-asparaginase or fed with asparagine-free diet had significantly reduced lung metastasis. This work provides evidence that environmental asparagine might be limiting in the lung or during the process of lung metastasis and thus render de novo biosynthesis to be essential. Mechanistically, asparagine restriction reduces the expression of genes involved in the epithelial-mesenchymal transition (EMT), a key step for metastasis to initiate [78]. Since EMT process happens at the primary sites, it still warrants further investigation on the relative amount of asparagine needed for primary tumor growth versus EMT process. 

The ability of asparagine to modulate tumor progression can also be reflected at the stage when tumor cells respond to therapeutics. A recent study (in preprint) shows that clonal heterogeneity within a tumor tissue can contribute to therapeutic resistance through asparagine biosynthesis [79]. In a mouse pancreatic tumor model, the authors discovered metabolic heterogeneity within a tumor tissue, which leads to differential responses to electron transfer chain inhibitors (ETCi). Of interest, clones that were sensitive to ETCi become resistant when they are co-cultured with clones that are intrinsically resistant to ETCi. Through comprehensive metabolic analysis, the manuscript identified asparagine as a key metabolite secreted from the resistant clones, which can feed the sensitive clones to drive their resistance to ETCi. Since a major tumor suppressive effect of ETCi is via restricting aspartate production [81,82], the manuscript found that asparagine supplementation can preserve aspartate pool in sensitive clones likely by diverging aspartate from asparagine biosynthesis [79]. Furthermore, L-asparaginase treatment sensitizes tumor cells to phenformin, a complex I inhibitor, in an allograft mouse pancreatic tumor model in vivo. 

Similar to findings in leukemic blasts, the synergy between L-asparaginase and WNT pathway activation can also be observed in solid tumors. Similar to ALL cells, the same group found that WNT activation by R-spondin fusion in colorectal cancers drives selective sensitivity to L-asparaginase treatment [80]. Of interest, APC mutations that activate beta-catenin downstream of GSK3 shows little response to L-asparaginase treatment. Mechanistically, APC mutant colorectal cancer cells retain high GSK3 activities to drive proteolytic scavenging of asparagine; in contrast, R-spondin fusion suppresses GSK3 activation to de-repress beta-catenin activity, and thus is unable to activate GSK3-dependent proteolysis [80]. Similar to ALL cells, it is unclear whether this GSK3-dependent sensitivity to L-asparaginase involves the regulation of ASNS expression or not.

Taken together, these work shows that asparagine plays a critical role as a key metabolite for tumor cell growth or survival at various stages during tumor progression. When environmental glutamine levels decline, asparagine can protect tumor cells from apoptosis or even support de novo glutamine biosynthesis in a context-dependent manner. In addition, genetic lesions in tumor cells can alter their dependencies on de novo biosynthesis or extracellular acquisition of asparagine, which provides rationale for targeted therapeutics. Furthermore, asparagine bioavailability can affect tumor progression at specific stages, including lung metastasis or response to treatment through stage-specific mechanisms. 

### 3.2. Asparagine Regulates Cancer Cell Signaling

In addition to its canonical role as an amino acid for protein synthesis, asparagine has been found to have non-metabolic roles in regulating tumor-associated signaling. These studies will provide not only a comprehensive picture into the molecular mechanisms by which asparagine facilitates tumor progression, but also insights into the potential reasons for therapeutic resistance following asparagine depletion.

The initial attention of asparagine in regulating cancer cell signaling was reported by a seminal work demonstrating that asparagine can activate mTORC1 to drive protein synthesis and nucleotide biosynthesis [83]. The ability of asparagine to activate mTORC1 relies at least partially on its ability to function as an antiporter exchange factor for the import other amino acids (Figure 3A). Recently, the same group found that the ability of asparagine to activate mTORC1 represents a key adaptive strategy for tumor cells to mitigate ETC inhibition-induced stress, which is coupled to ATF4 activation [84]. This work provides an alternative explanation for the synergy between ETC inhibition and asparagine restriction [79] (Figure 3A), and also brings up the question of whether asparagine or aspartate is the most limiting metabolite to drive ETC inhibition-associated phenomenon in tumor cells. Furthermore, in addition to functioning as an exchange factor for mTORC1 activation, asparagine can also activate mTORC1 directly through an ADP-ribosylation factor 1 (Arf1)-dependent but Rag GTPase-independent mechanism [85] (Figure 3A). 

Asparagine can also alter the activities of signaling molecules through direct binding. A recent work shows that asparagine can bind to LKB1, an upstream suppressor of the AMP-activated protein kinase (AMPK), to promote the inhibitory effect of LKB1 on AMPK [86] (Figure 3B). As a result, intracellular depletion of asparagine activates AMPK, which induces p53 phosphorylation to further suppress ASNS transcription through promoter recruitment. It remains to be determined why tumor cells use this feedforward loop to control the expression of ASNS when they suffer from asparagine limitation. In addition, it will be interesting to investigate whether p53 status can be used to predict sensitivity to asparagine depletion in a broader spectrum of tumors, considering its frequent loss of function in cancers. Of interest, the same group recently reported that asparagine can directly enhance the T cell receptor (TCR) signaling to promote CD8+ T cell activation and its anti-tumor responses [87]. Mechanistically, asparagine directly binds to lymphocyte-specific protein tyrosine kinase (LCK) to induce its autophosphorylation on Tyr 394 and 505, which consequently leads to enhanced TCR signaling (Figure 3B). More importantly, immunocompetent mice pre-fed with asparagine-free diet shows decreased activities of CD8+ T cells and compromised anti-tumor responses in a subcutaneous B16 melanoma model [87]. This study also highlights the potential challenge of using T cell-based therapy in cancer patients if they will be treated with L-asparaginase. 

ALL cells adapt to asparagine depletion through engaging the GCN2/eIF2α/ATF4 axis to turn on the expression of ASNS. However, the GCN2/eIF2α feedback loop seems not to be sufficient to induce the ATF4 in melanoma cells to mediate adaptation. Indeed, melanoma cells rely on GCN2/eIF2α axis to induce the expression of receptor tyrosine kinases (RTKs), which initiates a signaling cascade through MAPK/mTORC1/eIF4E axis to achieve maximal induction of ATF4 [88] (Figure 3C). This study demonstrates that MAPK inhibitors can synergize with L-asparaginase in a subcutaneous mouse melanoma model through preventing the ATF4 accumulation and ASNS induction. In addition, the same group reported recently that c-MYC is a key downstream component of the MAPK for mTORC1 activation [89]. This process is regulated through c-MYC-dependent transcription of amino acid transporters, such as SLC7A5. As a result, increased essential amino acid import triggers mTORC1 activation (Figure 3C). 

Taken together, these studies shed light into the signaling regulatory role of asparagine. Although mTORC1 can either be activated by asparagine or can be feedback turned on through RTKs/MAPK during asparagine starvation, its sustained activation is critical to couple amino acid availability to protein synthesis and other anabolic reactions. Asparagine depletion can also activate AMPK, an energy sensor in cells. This suggests that the lack of intracellular asparagine may confer a non-anabolic status to preserve energy to mitigate the stress responses. Furthermore, the fact that asparagine can facilitate CD8+ T cell activation and their anti-tumor responses will bring up the challenge of using L-asparaginase or asparagine-restricted diet in cancer patients concurrent with immunotherapy. 

## 4. Conclusions

In summary, asparagine is a key metabolite in supporting tumor progression. Elucidating the molecular mechanisms that drive resistance to L-asparaginase treatment in ALL will further advance our understanding of why other cancers do not respond to L-asparaginase treatment. As a result, combining L-asparaginase with novel targeted therapeutics overcoming resistance, will potentially broaden its application in other cancers, in the future. Further studies on immune-tumor interaction in specific tumor microenvironments are needed urgently to tailor potential therapeutic strategies to preserve anti-tumor immune responses while targeting resistant cancer.

## Figures and Tables

**Figure 1 metabolites-11-00402-f001:**
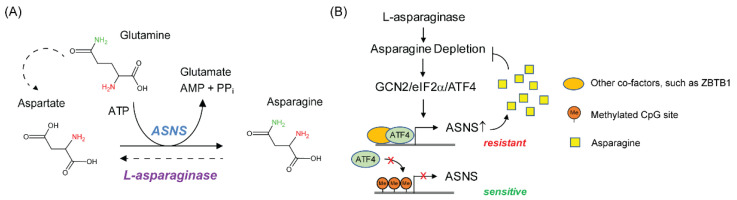
Asparagine de novo biosynthesis in driving L-asparaginase resistance. (**A**) ASNS catalyzes asparagine biosynthesis by using glutamine, aspartate and ATP. In proliferating mammalian cells, most aspartate is synthesized de novo by using oxaloacetate and glutamate as substrates. Since glutamate is produced through glutamine deamination, which can be further deaminated to fuel the TCA cycle to generate oxaloacetate, glutamine is the major carbon and nitrogen donor for aspartate biosynthesis [56]. Catabolism of asparagine to aspartate by L-asparaginase has not been reported in mammalian cells. (**B**) Asparagine depletion by L-asparaginase activates GCN2 pathway, leading to ATF4 accumulation, which turns on ASNS. As a result, cells synthesize more asparagine to mitigate the stress. However, ATF4 cannot be recruited to the ASNS promoter unless it is demethylated.

**Figure 2 metabolites-11-00402-f002:**
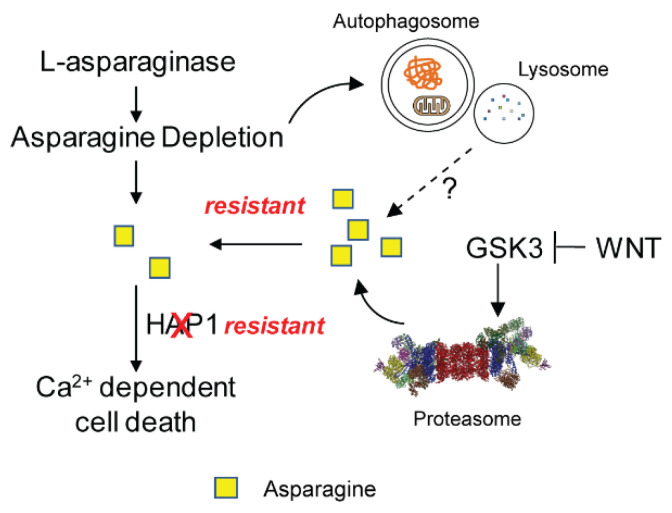
Mechanisms driving L-asparaginase resistance independently of ASNS. Potential contribution of proteolytic/autophagic scavenging of asparagine and Ca^2+^-dependent cell death inhibition to L-asparaginase resistance.

**Figure 3 metabolites-11-00402-f003:**
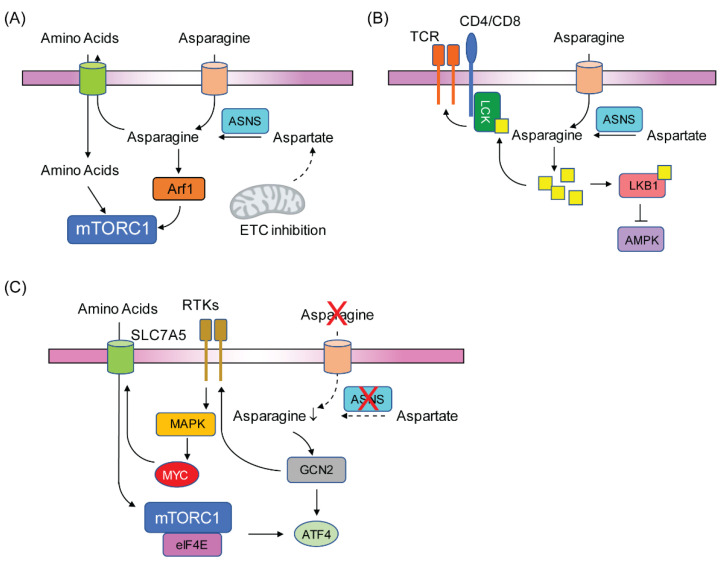
Signaling pathways regulated by asparagine or its depletion. (**A**) Asparagine activates mTORC1 through Arf1 or through importing other amino acids. Asparagine can also maintain mTORC1 activity during ETC inhibition. (**B**) Asparagine can directly bind to LCK or LKB1 to modulate their activities. As a result, asparagine is a positive regulator of TCR signaling and a negative regulator of AMPK pathway. (**C**) Asparagine depletion activates MAPK pathway through the induction of RTKs. MAPK activation leads to the engagement of MYC-SLC7A5 axis to support amino acid uptake, which subsequently activates mTORC1/eIF4e to support the translation of ATF4 protein.

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
