# Peer review of "Asparagine: A Metabolite to Be Targeted in Cancers"

_metabolites, 2021, doi:10.3390/metabo11060402_

Round 1
Reviewer 1 Report
This was a well written, short review. There is much more in the literature surrounding asparagine addicted cancers. This manuscript does a good job describing the limited aspect of asparagine in cancer selected. Since it is such a wide field of study, this is a nice piece of work that adds to the literature. The presumption about autophagy as bein an amino acid scavenger is misleading and needs to be edited. For other comments, please see the annotated PDF file.

Author Response
Thanks for the reviewer's suggestion.
- In line 106-108, we specified that the conclusion is made from the Pseudomonas 7A
- In line 166-170, we described how aspartate is synthesized from glutamine and included a reference.
- In line 211-212, we added two references. One is a review article, and the other is a research article. Both articles suggest that autophagy can function as an amino acid scavenging mechanism under nutrient limiting conditions.
- We separated the original Figure 1C as a new figure (Figure 2) between line 222 and 223.
- We moved the original Figure 2, now as revised Figure 3, to line 373.
Reviewer 2 Report
Asparagine is a fundamental metabolite in supporting tumour progression. The possibility of carrying out studies considering its use in several tumours development might offer a new tool with a great potential for an improved diagnosis and prognosis, leading to more personalized and effective patient treatments.
This article seems extremely clear to me and the topics were presented in a linear and nicely presented way.
Therefore, in my opinion, this paper should be published in the present form.
Provided below are some comments for the authors which may further improve the manuscript.
- Line 17 (abstract) literatures -> literature (it is an uncountable noun)
- Consider adding the hyphen(s) where needed (e.g. genome-wide)
- Authors should replace Figure 1 and 2 with higher resolution images (as they are presented now the images are grainy)
Author Response
Thanks for the reviewer's comments. We have made changes accordingly.
Reviewer 3 Report
The authors make a review about L-aspargine and ASCS in acute lymphoblastic lymphom, and other cancers. The review is written in very well structured way and easy to follow. I speacially like how the authors add a summary paragraph at the end of each section.
I just found something to correct: Third affiliation is just a department.
Author Response
Thanks for the reviewer's comments, we have made changes accordingly.